# Enhanced Mechanical and Thermal Properties of Modified Oil Palm Fiber-Reinforced Polypropylene Composite via Multi-Objective Optimization of In Situ Silica Sol-Gel Synthesis

**DOI:** 10.3390/polym13193338

**Published:** 2021-09-29

**Authors:** Nasrullah Mat Rozi, Hamidah Abdul Hamid, Md. Sohrab Hossain, Nor Afifah Khalil, Ahmad Naim Ahmad Yahaya, Ahmad Noor Syimir Fizal, Mohd Yusoff Haris, Norkhairi Ahmad, Muzafar Zulkifli

**Affiliations:** 1Green Chemistry and Sustainability Cluster, Branch Campus Malaysian Institute of Chemical and Bioengineering Technology, University Kuala Lumpur, Taboh Naning, 78000 Alor Gajah, Melaka, Malaysia; nasrullah5157@gmail.com (N.M.R.); hamidah.abdhamid@unikl.edu.my (H.A.H.); norafifah@unikl.edu.my (N.A.K.); ahmadnaim@unikl.edu.my (A.N.A.Y.); syimir.fizal@s.unikl.edu.my (A.N.S.F.); 2School of Industrial Technology, Universiti Sains Malaysia, 11800 USM, Penang, Malaysia; 3Aerocomposite Cluster, Branch Campus Malaysian Institute of Aviation Technology, University Kuala Lumpur, 43900 Sepang, Selangor, Malaysia; mohdyusoff@unikl.edu.my; 4Industrial Linkages Section, Branch Campus Malaysian France Institute, Universiti Kuala Lumpur, 43650 Bandar Baru Bangi, Selangor, Malaysia; norkhairi@unikl.edu.my

**Keywords:** oil palm fiber, silica sol-gel, natural fiber reinforced composite, polymeric composite, polypropylene, maleic anhydride grafted polypropylene

## Abstract

A multi-objective optimization of in situ sol-gel process was conducted in preparing oil palm fiber-reinforced polypropylene (OPF-PP) composite for an enhancement of mechanical and thermal properties. Tetraethyl orthosilicate (TEOS) and butylamine were used as precursors and catalysts for the sol-gel process. The face-centered central composite design (FCCD) experiments coupled with response surface methodology (RSM) has been utilized to optimize in situ silica sol-gel process. The optimization process showed that the drying time after the in-situ silica sol-gel process was the most influential factor on silica content, while the molar ratio of TEOS to water gave the most significant effect on silica residue. The maximum silica content of 34.1% and the silica residue of 35.9% were achieved under optimum conditions of 21.3 h soaking time, 50 min drying time, pH value of 9.26, and 1:4 molar ratio of TEOS to water. The untreated oil palm fiber (OPF) and silica sol-gel modified OPF (SiO_2_-OPF) were used as the reinforcing fibers, with PP as a matrix and maleic anhydride grafted polypropylene (MAgPP) as a compatibilizer for the fiber-reinforced PP matrix (SiO_2_-OPF-PP-MAgPP) composites preparation. The mechanical and thermal properties of OPF-PP, SiO_2_-OPF-PP, SiO_2_-OPF-PP-MAgPP composites, and pure PP were determined. It was found that the OPF-S-PP-MAgPP composite had the highest toughness and stiffness with values of tensile strength, Young’s modulus, and elongation at break of 30.9 MPa, 881.8 MPa, and 15.1%, respectively. The thermal properties analyses revealed that the OPF-S-PP-MAgPP exhibited the highest thermally stable inflection point at 477 °C as compared to pure PP and other composites formulations. The finding of the present study showed that the SiO_2_-OPF had the potential to use as a reinforcing agent to enhance the thermal-mechanical properties of the composites.

## 1. Introduction

Natural fibers are gaining significant attention for their potential in replacing the conventional synthetic fibers for the fiber reinforcement into polymeric composites in terms of environmental benefits, high stiffness-to-weight ratio, and relatively inexpensive feed stocks [1,2,3]. Recently, there is an increasing interest on natural fibers derived from biomass to produce natural fiber-reinforced composites (NFRCs) [4,5,6]. The structure of biomass consists of rigid crystalline cellulose in a soft amorphous matrix of lignin and hemicellulose, which also contains pectins, waxes, and water-soluble substances [7,8]. Biomass fibers are easily available and serve as low-cost reinforcement materials to the polymeric composites, which could provide a profitable and sustainable solution to the biomass disposal issue. Oil palm fiber (OPF) is one of the major contributors to the global biomass fibers generation, which consists of huge amounts of empty fruit bunches, mesocarp fibers, palm kernel shells, oil palm trunks, and oil palm fronds [9,10,11]. It is estimated that worldwide generation of oil palm fibers would be 149 million ton in 2020, which is twofold of a total mass of extracted palm oil [12]. Moreover, the OPF which contains 48 to 65% cellulose has specific modulus and high specific strength comparable to glass fibers [13,14,15].

There are some processing issues of NFRCs, such as hygroscopic properties of natural fiber that leads to the dimensional instability of NFRCs, surface fiber degradation at temperature above 200 °C, and the complexity of composites machining [1,16,17,18]. Nevertheless, the limitations of reinforcing the natural fibers into polymer matrices could be improved by a pre-processing step, such as modifications of the polymer matrices and surfaces of natural fiber via physical or chemical technologies [19,20,21]. The use of thermoplastic polymers such as polypropylene (PP) and polyethylene [22,23] to bind the reinforcing fibers has less difficult processing technique and better design flexibility than thermosetting polymers such as epoxy and phenolic [15,24]. The development of OPF-reinforced PP composite has high potential to be used for light-weight automotive industry components, due to its improvement in tensile modulus, flexural strength, and flexural modulus [25]. However, the incompatibility of OPF-PP composite would cause a poor interfacial adhesion between the reinforcing OPF and PP matrix [26]. Hence, an addition of a coupling agent such as maleic anhydride-grafted polypropylene (MAgPP) during a melt blending would improve its impact resistance, stiffness, and dimensional stability [23,27,28]. Moreover, the OPF residual oil removal and alkali treatment to eliminate the impurities on their surfaces, result in better adhesion between OPF and the polymer matrix [22].

Furthermore, OPF has leakage concern due to its poor thermal stability, which limits its applications. A formation of gel-like diphasic colloidal solution (sol-gel) is an attractive low-cost technology to enhance the thermal and structural performance of OPF [29]. Tetraethyl orthosilicate (TEOS) is often used in silicon dioxide (silica) sol-gel synthesis due its low toxicity. The utilization of TEOS as a precursor to the sol-gel process has improved the properties of OPF in terms of enhanced mechanical properties, thermal stability, and melting point [30,31,32]. Moreover, an in situ silica sol-gel has been used as an effective filler to reduce the void content of lignocellulosic fibers, which favor high performance polymeric composites [24,33]. During a complete hydrolysis of TEOS, a network formation of hydrolyzed monomers and siloxane bonds (Si-O-Si) would occur and breaks down to produce silica sol-gel as the main product [34]. A complete in situ synthesis of silica sol-gel could be achieved by using an excess water and/or the use of chemical catalysts such as butylamine, fluoride, and ammonium hydroxide [32,34]. High molar ratio of water to TEOS (≥4) would be sufficient to drive toward a complete conversion of silica [35]. Instead, an early stage of partial reversible hydrolysis could occur, leading to alcohol and water condensations to form siloxane bonds (Figure 1).

Studies have been conducted on the synthesis of natural fiber-reinforced thermoplastics composite [15,19,25]. However, there are rarely any studies that have been conducted on the synthesis of polymeric composite reinforced with modified OPF with sol-gel silica. Utilization of silica produced via sol-gel process for lignocellulosic fibre in thermoplastics showed that the dispersion of silica nano particle size 10–50 nm through sol-gel in the matrix shows improvement in the tensile modulus and thermal stability of LDPE-based composites [35]. It is also stated that the sol-gel process improves the dispersion of filler and adhesion to hydrophobic matrix such as polyethylene and reaction speed is controllable. A biodegradable composite of PLA/wood flour (WF)/SiO_2_ was also developed by utilizing TEOS via the sol-gel process [36]. The composite produces tensile strength improvement of 20 MPa and initial decomposition temperature (IDT) increased with 10 wt. % of silica present in the composite. This increases greatly the thermal stability of the composite. Therefore, in the present study, butylamine and TEOS were used as alkaline catalyst and precursor respectively for in situ silica sol-gel synthesis. The synthesis was optimized by using a response surface methodology combined with face-centered central composite design (FCCD). The optimization of in situ silica sol-gel process is essential as an initial process that enhances the mechanical and thermal properties of OPF-S-PP-MAgPP composite. The comparison of mechanical and thermal properties between the investigated composites was also carried out.

## 2. Materials and Methods

### 2.1. Materials

The OPF was supplied by Kilang Sawit Meru Sdn. Bhd. (Meru Oil Palm Plant, Private Limited) (Shah Alam, Malaysia) and the PP (PP456J, extrusion grade) was purchased from Lyondell Basell Industries N. V. (Rotterdam, The Netherlands). Tetraethyl orthosilicate (TEOS), butylamine, and hydrochloric acid (HCl) were acquired from Merck (Shah Alam, Malaysia) while maleic anhydride-grafted polypropylene (MAgPP) was purchased from Lotte Chemical Titan (M) Sdn. Bhd. (Johor Bahru, Malaysia).

### 2.2. Design of Experiment and Optimization of In Situ Silica Sol-Gel Synthesis

Beforehand, OPF was dried in a conventional air flow oven (Memmert Drying Oven ULE 500, Schwabach, Germany) at a temperature of 105 °C for 48 h until a moisture content of less than 5% was achieved. The dried OPF was then ground and sieved to obtain the desired range of particle sizes between 100 and 300 μm. Response surface methodology combined with FCCD was used to conduct an optimization study of an in situ silica sol-gel synthesis on OPF. Design Expert (Version 11: Minneapolis, MN, USA) was used to run a matrix of 31 experimental runs and to analyze the results. The ranges of four independent variables (factors) for the silica sol-gel process are soaking time, drying time, pH value and molar ratio of TEOS to water, as tabulated in Table 1.

The samples of OPF were first soaked into butylamine solution with 2.5% purity and a pH value of 11 for 24 h, to clean the external surface and enlarge the pores of OPF samples, so that the silica sol-gel could penetrate inside the OPF effectively. The samples were then oven-dried for 24 h at 110 °C. The dried OPF samples were soaked into TEOS at specified molar ratio of TEOS to water. The treated OPF was soaked again in the butylamine solution for 1 h, at varied pH values and final soaking times. The pH value of solution was controlled and stabilized by using a titration of butylamine solution and HCl solution. Then, the OPF samples were oven-dried at varied drying times. The investigated responses are the silica content determined by using Equation (1) and the amount of silica residue from the thermogravimetric analysis.
(1)$$Silica\;content\;\left(SC\%\right)=\frac{w2-w1}{w1}\times \;100$$
where *w*1 is a weight of OPF in g unit and *w*2 is a weight of modified silica sol-gel OPF (OPF-S) in g unit. Meanwhile, the amount of silica residue from thermogravimetric analysis was determined after the silica sol-gel processing was completed. The schematic diagram for the production of OPF and OPF-S conducted through sol-gel process is presented in Figure 2. The experiments were conducted in triplicate and the results are represented as the mean values from the triplicate experimental runs.

### 2.3. Preparation of Natural Fiber Reinforced Composites

The silica sol-gel modified-OPF based composite was prepared by a melt blend process using a twin-screw extruder (Model Haake™ P 300, Thermo Fisher Scientific, Waltham, MA, USA) by mixing 30% of untreated OPF and 70% of PP. Then, the blended sample was crushed into smaller pellets with a cruncher to provide larger contact areas. Then, the final stage was carried out by using an injection molding machine (Wittmann Battenfeld 60 tonne, Vienna, Austria). The preparation steps were repeated by using silica sol-gel modified-OPF and PP-MAgPP to fabricate silica sol-gel modified-OPF-PP and silica sol-gel modified-OPF-PP-MAgPP composites. The compounding formulation of composites are depicted in Table 2.

### 2.4. Mechanical Characterization

Tensile and flexural tests of 7 specimens from each type of composite were performed according to ASTM D3039 and ASTM D790, respectively using Lloyd universal testing machine (honglin, Shandong, China)with a strain rate of 50 mm/min at a load of 1 kN. Meanwhile, an impact test for 7 specimens from each type of composite was conducted using Ray-Ran Pendulum Impact Tester (Selangor, Malaysia) according to ISO 179, with a load of 1.189 kg and velocity rate at 2.9 m s^−1^. All samples were conditioned for 24 h at 23 ± 2 °C and 50 ± 5% relative humidity (RH) before the tests were carried out.

### 2.5. Termogravimetric Analysis (TGA)

The thermal degradation of the composite samples was tested using ASTM E1131 and analyzed using Mettler Toledo TGA machine (Selangor, Malaysia), with complete nitrogen and air burning conditions. The samples were weighed at 7–15 mg at constant heat rate of 10 °C/min to 800 °C with nitrogen and 800 to 1000 °C with air purge at 50 mL/min. The curve and maximum inflection point were analyzed to determine the thermal stability of the samples.

### 2.6. Surface Morphology Analyses

The broken impact testings of the composites were carried out using ultra high-resolution scanning electron microscope (FESEM) Hitachi-SU 8000 (Tokyo, Japan). The analysis studies the morphology of the composite, the content of silica on the surface and inside the OPF, and determines the size of the void inside of the OPF. Platinum was used to coat the samples to make them more conductive. The analysis was then supplemented with elemental analysis (EDX). The size of the void is determined by using Image J software (Tokyo, Japan). The samples were first coated with a platinum coating to make them more conductive. An elemental analysis at the end was also vital in determining the percentage of components involved in the measured area for identification purposes. About 50 measurements were recorded for each condition used for the SEM images representation.

## 3. Results and Discussion

### 3.1. Multi-Objective Optimization of In Situ Silica Sol-Gel Synthesis

The experimental sequence of in situ silica sol-gel synthesis on OPF was randomized to minimize the experimental runs and effects of extraneous factors. The actual design of experiments (DOE) and results of both responses, which were silica content and silica residue from TGA are presented in Table 3.

These results were fitted in quadratic polynomial models by applying multiple regression analysis:SC (%) = −485.1 + 70.26 A+ 17.06 B − 0.265 C + 8.03 D − 0.1905 AB + 0.00192 AC + 0.078 AD − 0.00344 BC − 0.184 BD − 0.0829 CD − 3.645 A^2^ − 0.325 B^2^ + 0.003429C^2^ + 0.902 D2(2)
SR (%) = 332.1 + 74.09 A + 0.99 B − 0.032 C + 16.78 D − 0.0024 AB + 0.00085 AC − 0.282 AD + 0.00198 BC − 0.786 BD − 0.03744 CD − 4.082 A^2^ + 0.0149 B^2^ + 0.000594 C^2^ + 1.459 D^2^(3)
where the responses, which are SC (%) and SR (%) represent silica content, silica residue from TGA result after the completion of silica sol-gel synthesis (%) respectively. Meanwhile, A, B, C, and D denote four investigated independent variables, which are pH value, soaking time (h), drying time (min), and molar ratio of TEOS to water, respectively.

The model represented by Equation (2) infers that an increase in silica content is due to increases in pH value and soaking time, while reductions in drying time and molar ratio of TEOS to water decreases the silica formation. On the other hand, based on Equation (3), an increase in pH value and reductions in molar ratio of TEOS to water, soaking and drying times would increase the silica residue from TGA. The analysis of variance (ANOVA) results for both silica content and silica residue are shown in Table 4 and Table 5, respectively. Overall, the probability values (*p*-values) of less than 0.05, imply that the regression model was statistically significant for predicting both silica content and silica residue and therefore the models are utilized to elucidate all independent variables precisely. Besides, regression coefficient (*R^2^*) values for percentage silica content and silica residue were 0.9354 and 0.9508, respectively. Wherein, the adjusted regression coefficient (*R^2^Adj*) for the percentage silica content and silica residue were 0.8789 and 0.9077, respectively. Both R^2^ and R^2^Adj values reveals that the regression model was good fitted to the experimental data for both silica content and silica residue. The insignificant lack of fit implies that the proposed regression model was sufficient to describe the relationship of independent variables and responses for analyzing both silica content and silica residue.

For the single-factor effect, the pH and the molar ratio of TEOS to water were statistically significant to the percentage silica content. However, the molar ratio of TEOS to water was the only independent variable that showed a strong significance to predict the silica residue from TGA. When pH value, soaking time, and the molar ratio of TEOS to water were kept constant at design center-points (9, 22 h, and 1:2.5 respectively), it was observed that the drying time of treated OPF gradually increased the silica content, as illustrated in Figure 3a. However, a reduction of silica content was observed when the sample was dried at more than 60 min. By using oven-drying method, high volumetric shrinkage and highly dense structure of sample could be obtained [37]. Nevertheless, since the alkali-catalyzed hydrolysis of TEOS would yield highly branched silica oligomers [34], it is suggested that a prolonged drying time of more than 60 min would produce a smaller pore size of OPF than the silica sol-gel. Hence, the forced removal of the silica network from the voids of OPF would occur. On the other hand, silica content and its residue from TGA results were inversely proportional to the molar ratio of TEOS to water when pH value, soaking and drying times were maintained at 9, 22 h, and 75 min (Figure 3b,c). The highest silica sol-gel formation was achieved at 1:4 molar ratio of TEOS to water, which was evidently sufficient in yielding more silica sol-gel into OPF [38].

Figure 4 shows the response surface plots of main interactive effects of variables on silica sol-gel synthesis from tetraethyl orthosilicate (TEOS). The three-dimensional plot in Figure 4a signifies that the silica content at pH value of 9 is the most sensitive with changes in molar ratio of TEOS to water. However, a minimal effect of molar ratio of TEOS to water on silica content was observed at lowest and highest pH values. The hydrolysis of TEOS with an excess water would favor a three-dimensional framework of siloxane bonds formation, which finally leads to the silica sol-gel process [39]. The result suggests that the maximum formation of silica sol-gel could be achieved at the lowest molar ratio of TEOS to water and pH values between 8 and 10. On the other hand, a moderate effect of soaking time-drying time on silica residue was observed. The highest silica residue was obtained within a middle contour, which fell in the ranges between 60 and 80 min drying times and 20 and 24 h soaking times. By using a suitable range of soaking time, the hydrogen bonding between butylamine molecules and OPF results in a formation of a thin layer, which could prevent hydroxyl groups of OPF from participating in the hydrolysis of TEOS to silica [40].

The criterion selected in this multi-objective optimization was to maximize the silica content and silica residue from TGA. Design expert software (ver.11) was utilized to optimize the experimental conditions of pH, soaking and drying times, and molar ratio of TEOS to water to maximize the silica content and silica residue. Three independent variables such as pH value, soaking and drying times were set within the design ranges, while molar ratio of TEOS to water was set at the lowest value as possible. Based on the stipulated goals, the predicted optimum silica content and silica residue within 95% confidence interval were 37.82% and 36.49% respectively. The optimum conditions obtained for the silica sol-gel process were pH value of 9.26, 21.3 h soaking time, 50 min drying time and 1:4 molar ratio of TEOS to water of 1:4, which was based on the highest desirability of 0.796. After experimental triplication and verification, the optimum values silica content and silica residue were obtained at 34.1% and 35.9%, with their standard deviations of 8.1 and 8.0, respectively.

Silica content was observed in the formed OPF-S with a burn test at 30.2 wt.%. A 1.2% residue of OPF was found after the completion of burning at 600 °C for 6 h. The untreated OPF burned off at 98.8%. During the in situ sol-gel silica process modification of OPF, there was a reaction between OPF and butylamine molecules via hydrogen bond, which formed a thin layer. This layer then prevents the hydroxyl groups of OPF from taking part in the reaction with TEOS [40]. This can be visualized in the OFP and OFP-S SEM images, which are shown in Figure 5. The SEM image of the OPF shows the presence of voids and the absence of silica on the surface (Figure 5a). The surface of the OPF-S clearly shows the presence of silica particles partially covering the surface and the voids. The range of silica particle size is from 01.5 micron to 0.20 micron in diameter.

Figure 6 shows the elemental analysis of OPF and OPF-S with EDX. Both readings were conducted in order to study the void area present on the lignocellulosic fiber. OFP was found with oxygen (O), carbon (C), and platinum (Pt) present. Elemental analysis of OFP-S revealed the presence of O, C, Pt and silica (Si). Si was found with high concentration in the focus area compared to other components. From these findings of EDX and SEM results, the presence of silica in the voids areas of the OPF-S and surrounding surface was proven.

### 3.2. Mechanical Properties

Generally, the maximum load that samples could support without fracture while being stretched was measured as tensile strength. A modulus of elasticity by dividing the longitudinal stress over strain is known as Young’s modulus, while elongation of break determines the cracking resistance of samples upon stretching and changing in shape. The tensile properties of PP, untreated OPF-PP, modify OPF-S-PP and OPF-S-PP-MAgPP and are shown in Figure 7. It was observed that OPF as the reinforcing natural fiber has a significant role in increasing the value of Young’s modulus. The presence of silica increases the polarity of the composite from 34.1% to 35.9% during the chemical interactions between OPF-S and PP [41]. On the contrary, the elongation of break for all composites were reduce by 5% to 10% than the pure PP, while an improvement in tensile strength was observed in the presence of silica sol-gel and MAgPP in the composite structure. The presence of silica at 55% could reduce the void content in OPF surface, which results in less porosity of the composite structure. The same trends of tensile strength and Young’s modulus were also reported by using K-carrageenan-silica composite [42]. Furthermore, the addition of MAgPP as a coupling agent or compatibilizer improves interfacial bonding between natural fiber and PP matrix [43], which enhances the tensile strength of the composite. Hence, it is suggested that OPF-S-PP-MAgPP composite showed the highest toughness and stiffness with values of tensile strength, Young’s modulus, and elongation at break of 30.9 MPa, 881.8 MPa, and 15.1%, respectively.

Figure 8 illustrates the flexural properties and impact strength of PP and the examined composites. The flexural strength represents the required amount of force to bend the composite, while flexural modulus is related to the amount of force required to deform the composite. It was noted that high force was required to bend and deform the pure PP as compared to the examined composites. The result suggests that the silica might not be homogeneously spread and agglomerated on the surface of OPF, which would cause low stress transfer and therefore low flexural strength and flexural modulus [23]. Nevertheless, the presence of silica sol-gel in OPF slightly improved the flexural strengths of OPF-S-PP and OPF-S-PP-MAgPP composites at 17.8 MPa, compared to the flexural strength of the untreated OPF-PP composite at 16.7 MPa. Furthermore, all samples of OPF-reinforced PP composites displayed low values of impact strength, as compared to PP that has the highest value of impact strength at 6.9 kJ/m^2^. The result was due to the natural properties of polar (OPF and silica) and non-polar (PP) materials, which resulted in low tenacity to absorb mechanical energy under impact loading before fracturing. As aforementioned, the inhomogeneous spread of silica would lead to a lack of interfacial adhesion between OPF and OPF-S with PP, rendering the composite samples to be susceptible to fracture under the impact loading. Furthermore, the addition of MAgPP resulted in a decrease of impact strength caused by the inability of compatibilizer to improve interfacial adhesion between the reinforcing natural fiber and polymer matrix [44].

### 3.3. Thermal Decomposition Analysis

Figure 9 presents the TGA curves of different composite formulations. At the first stage, OPF-S-PP and OPF-S-PP-MAgPP specimens thermally degraded at slightly lower temperature than OPF-PP, due to the condensation of ≡Si−OH groups. All composites displayed similar trends of rapid exothermic weight loss at the second stage, which would be the oxidation of carbon functional groups. OPF-S-PP and OPF-S-PP-MAgPP had a weight loss of 30% at a decomposition temperature of 461 °C (Table 6), which was 5 °C higher than OPF-PP. Successful penetration of silica sol-gel into the voids of OPF results in increased stability against thermal decomposition. OPF-S-PP-MAgPP showed the highest inflection point at 477 °C at the greatest change on the weight loss curve. This indicates the positive effects of the reinforcing OPF, silica sol-gel, and MAgPP coupling agent on thermal resistance. At temperature above 500 °C, the weight residue of OPF-PP approached zero. On the other hand, the weight residue of OPF-S-PP and OPF-S-PP-MAgPP were 5.6 and 5.7% respectively at decomposition temperature of 1000 °C. The resistance against thermal decomposition could also be contributed by additional heat capacity of silica based on similar observation previously reported [45].

## 4. Conclusions

In situ silica sol-gel process on OPF was an effective method to fill the voids of OPF, which enhanced the mechanical and thermal properties of oil palm fiber-reinforced polypropylene composite.The optimum silica content of 34.1% and silica residue of 35.9% were achieved under optimum conditions of 21.3 h soaking time, 50 min drying time, pH value of 9.26, and 1:4 molar ratio of TEOS to water. The drying time after in situ silica sol-gel process on OPF was the most significant independent variable on silica content, while silica residue from TGA was highly influenced by molar ratio of TEOS to water.MAgPP was introduced for better interfacial adhesion between OPF-S and PP. The tensile strength of OPF-S-PP-MAgPP was significantly improved at 30.9 MPa, but other mechanical properties showed insignificant differences as compared to OPF-S-PP.The OPF-S-PP-MAgPP showed the highest thermal stability with the highest inflection point of 477 °C. However, the degradation difference between OPF-S-PP and OPF-S-PP-MAgPP was marginal.The outcomes of this study will be useful for future works on formulations of modified oil palm fiber-reinforced polypropylene composite in many potential applications.

## Figures and Tables

**Figure 1 polymers-13-03338-f001:**
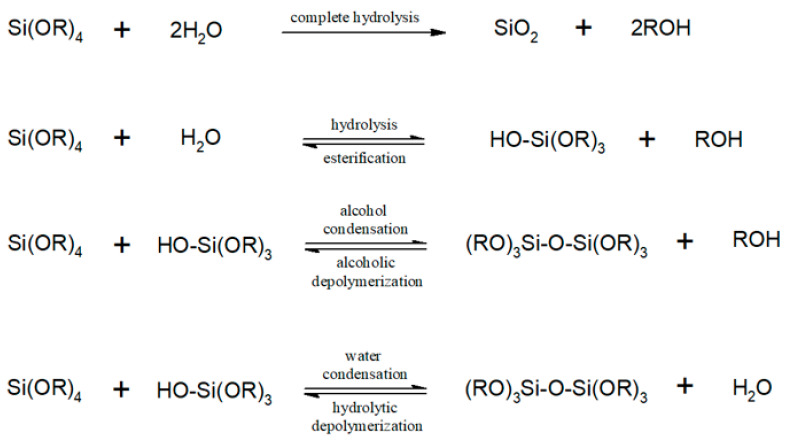
Hydrolysis, series of condensations and reverse reactions of silicon alkoxide.

**Figure 2 polymers-13-03338-f002:**
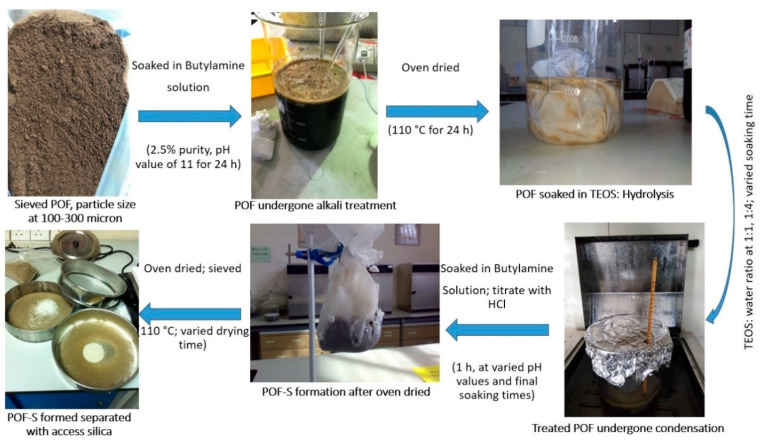
Schematic diagram of the production of OPF and OPF-S conducted through sol-gel process.

**Figure 3 polymers-13-03338-f003:**
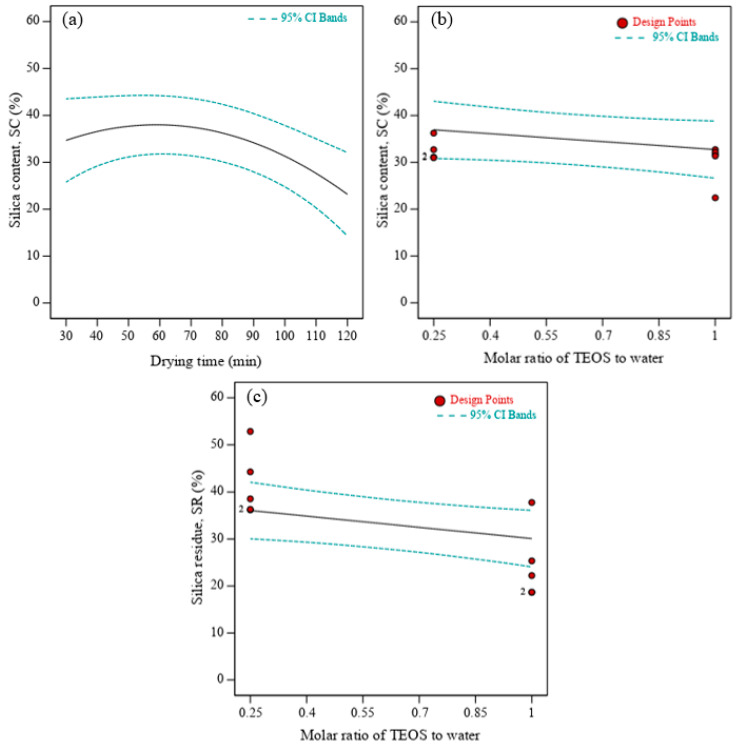
Single-effect of (**a**) drying time and (**b**) molar ratio of tetraethyl orthosilicate (TEOS) on silica content and (**c**) molar ratio of TEOS to water on silica residue.

**Figure 4 polymers-13-03338-f004:**
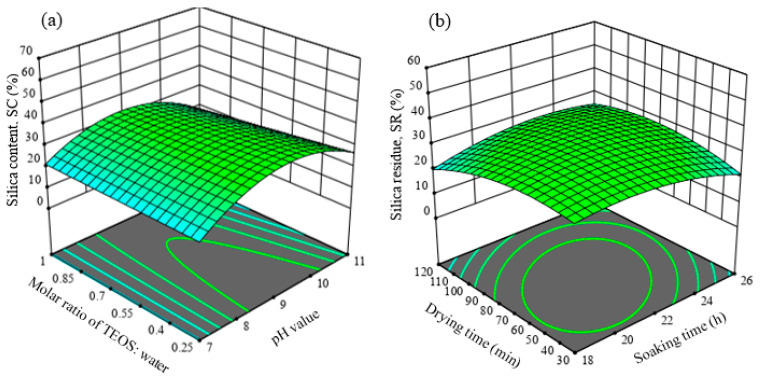
Response surface plots of main interactive effects of variables on silica sol-gel synthesis from tetraethyl orthosilicate (TEOS). (**a**) Interaction effect between pH and molar ratio of TEOS to water on silica content (%); and (**b**) Interaction effect between drying time and soaking time on silica residue (%).

**Figure 5 polymers-13-03338-f005:**
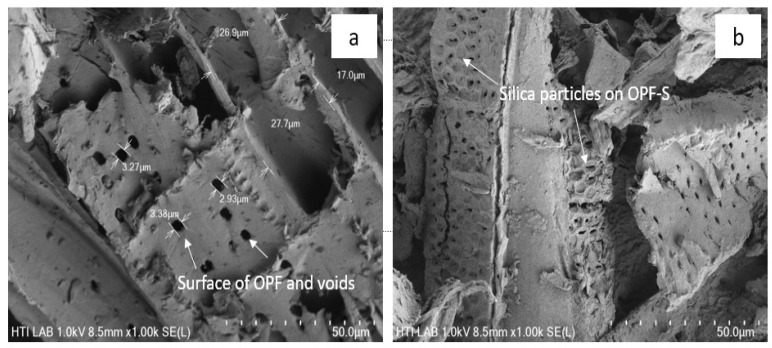
SEM images of the OPF (**a**) and OPF-S (**b**).

**Figure 6 polymers-13-03338-f006:**
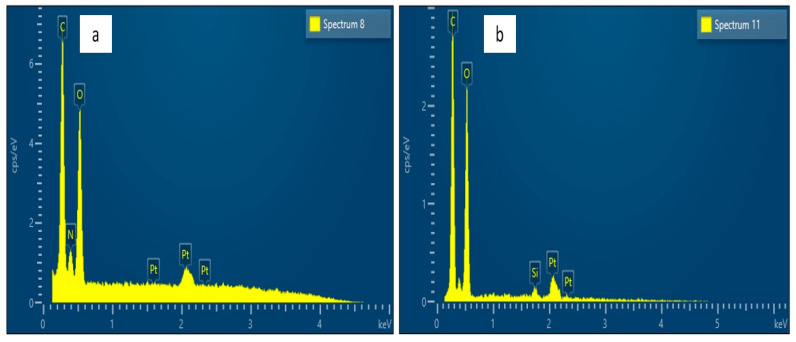
Elemental analysis of the OPF (**a**) and OPF-S (**b**).

**Figure 7 polymers-13-03338-f007:**
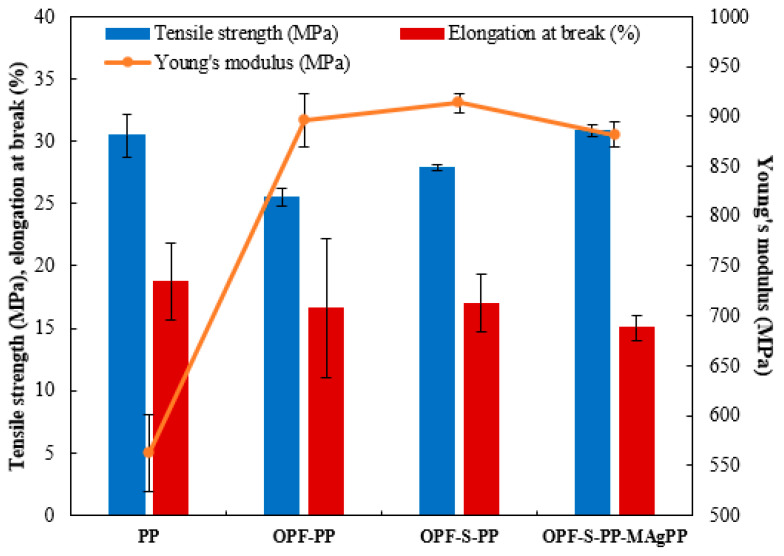
Tensile properties of PP, OPF-PP, S-OPF-PP, and S-OPF-PP-MAgPP composites.

**Figure 8 polymers-13-03338-f008:**
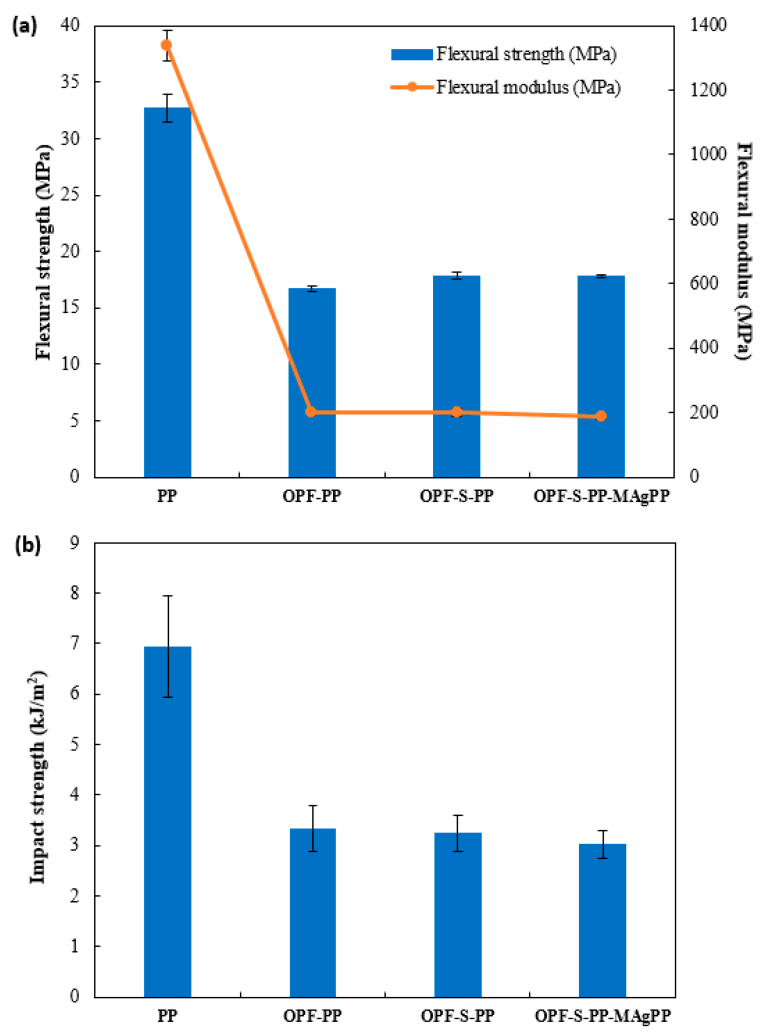
(**a**) Flexural properties and (**b**) impact strengths of PP, OPF-PP, OPF-S-PP, and OPF-S-PP-MAgPP composites.

**Figure 9 polymers-13-03338-f009:**
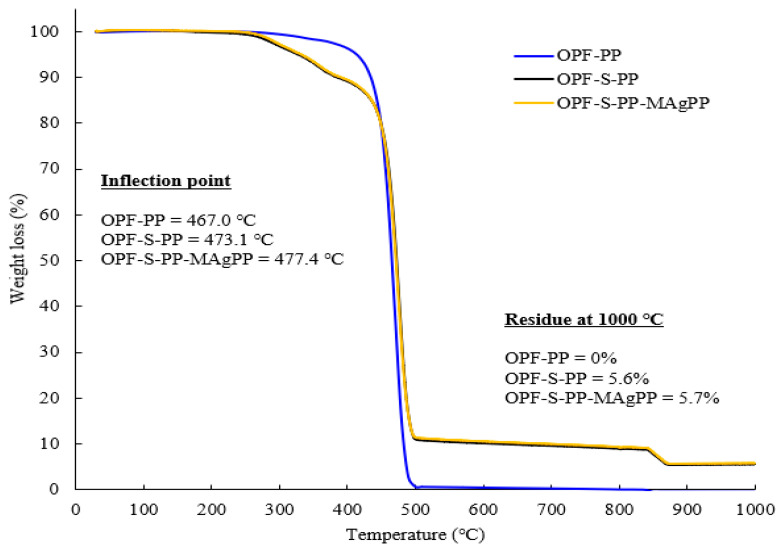
Thermogravimetric curves of OPF-PP, OPF-S-PP, and OPF-S-PP-MAgPP composites.

**Table 1 polymers-13-03338-t001:** Factors and ranges for in situ silica sol-gel silica synthesis on oil palm fiber.

Factor	Symbol	Levels		
		Low (−1)	Medium (0)	High (+1)
pH	A	7	9	11
Soaking time (h)	B	18	22	26
Drying time (min)	C	30	75	120
Molar ratio of TEOS/water	D	1:1	1:2.5	1.:4

**Table 2 polymers-13-03338-t002:** Compounding formulation of composites.

Sample	Polypropylene, PP (wt%)	Oil Palm Fiber, OPF (wt%)	Silica Sol-Gel Modified-OPF (wt%)	MAgPP Coupling Agent (wt%)
OPF-PP	70	30	-	-
SiO_2_-OPF-PP	70	-	30	-
SiO_2_-OPF-PP-MAgPP	70	-	30	10

**Table 3 polymers-13-03338-t003:** Central composite design of experiments for in situ silica sol-gel synthesis on oil palm fiber.

Run	A	B	C	D	Response			
					SC (%)		SR (%)	
					Actual	Predicted	Actual	Predicted
1	7	18	30	1:1	6.78	4.23	9.28	10.76
2	11	18	30	1:1	7.97	8.67	13.06	12.04
3	7	26	30	1:1	8.93	12.25	17.98	17.99
4	11	26	30	1:1	10.17	11.59	18.35	19.20
5	7	18	120	1:1	11.78	13.84	15.49	16.28
6	11	18	120	1:1	18.66	19.97	17.37	17.87
7	7	26	120	1:1	21.3	20.38	25.64	24.94
8	11	26	120	1:1	22.43	20.41	25.37	26.45
9	7	18	30	1:4	22.43	25.10	32.78	31.23
10	11	18	30	1:4	31.72	31.47	26.95	29.13
11	7	26	30	1:4	32.17	29.69	18.63	19.61
12	11	26	30	1:4	31.38	29.97	18.68	17.43
13	7	18	120	1:4	15.92	13.33	26.02	26.65
14	11	18	120	1:4	23.06	20.39	25.33	24.85
15	7	26	120	1:4	15.5	15.45	15.89	16.45
16	11	26	120	1:4	14.04	16.42	14.57	14.57
17	7	22	75	1:2.5	11.3	12.84	18.07	15.88
18	11	22	75	1:2.5	15.5	16.04	17.46	15.59
19	9	18	75	1:2.5	20.22	22.55	35.59	33.06
20	9	26	75	1:2.5	25.33	25.08	33.07	31.54
21	9	22	30	1:2.5	37.13	36.70	34.28	32.60
22	9	22	120	1:2.5	32.72	35.23	36.31	33.93
23	9	22	75	1:1	28.91	26.58	36.18	33.19
24	9	22	75	1:4	31.11	35.52	38.57	37.49
25	9	22	75	1:2.5	31.21	29.00	32.85	32.10
26	9	22	75	1:2.5	28.75	29.00	30.66	32.10
27	9	22	75	1:2.5	25.50	29.00	28.59	32.10
28	9	22	75	1:2.5	29.67	29.00	27.85	32.10
29	9	22	75	1:2.5	32.54	29.00	31.28	32.10
30	9	22	75	1:2.5	32.59	29.00	29.18	32.10
31	9	22	75	1:2.5	30.69	29.00	31.86	32.10

**Table 4 polymers-13-03338-t004:** Analysis of variance (ANOVA) for the percentage silica content.

Source	Sum of Squares	Degree of Freedom	Mean Square	*F* Value	*p*-Value
Model	2241.06	14	160.076	16.55	0.000
A-pH value	46.41	1	46.144	4.77	0.044
B-Soaking time	28.65	1	28.652	2.96	0.104
C-Drying time	9.78	1	9.783	1.01	0.329
D-Molar ratio of TEOS to water	359.12	1	359.120	37.14	0.000
AB	37.15	1	37.149	3.84	0.068
AC	0.48	1	0.476	0.05	0.827
AD	0.87	1	0.874	0.09	0.768
BC	6.13	1	6.126	0.63	0.438
BD	19.54	1	19.536	2.02	0.174
CD	500.64	1	500.64	51.77	0.000
A^2^	551.69	1	551.691	57.05	0.000
B^2^	70.32	1	70.316	7.27	0.016
C^2^	125.16	1	125.162	12.94	0.002
D^2^	10.69	0	10.691	1.11	0.309
Residual	154.73	17	9.670		
Lack of fit	118.10	10	11.810	1.93	0.217
Pure error	36.63	8	6.105		
Total	2395.79	31			
*R*^2^*= 0.9354*; *R*^2^*_Adj_ = 0.8789*					

**Table 5 polymers-13-03338-t005:** Analysis of variance (ANOVA) for the percentage silica residue from thermogravimetric analysis.

Source	Sum of Squares	Degree of Freedom	Mean Square	*F* Value	*p*-Value
Model	1878.43	14	134.173	22.07	0.000
A-pH value	0.39	1	0.387	0.06	0.804
B-Soaking time	10.41	1	10.412	1.71	0.209
C-Drying time	8.00	1	8.000	1.32	0.268
D-Molar ratio of TEOS to water	83.20	1	83.205	13.69	0.002
AB	0.01	1	0.006	0.00	0.975
AC	0.09	1	0.095	0.02	0.902
AD	11.48	1	11.475	1.89	0.188
BC	2.02	1	2.024	0.33	0.572
BD	355.79	1	355.794	58.52	0.000
CD	102.16	1	102.162	16.80	0.001
A^2^	691.82	1	691.821	113.79	0.000
B^2^	0.15	1	0.147	0.02	0.879
C^2^	3.75	1	3.754	0.62	0.443
D^2^	27.97		27.965	4.60	0.048
Residual	97.28	17	6.080		
Lack of fit	79.67	10	7.967	2.72	0.117
Pure error	17.60	6	2.934		
Total	1975.71	31			
*R*^2^*= 0.9508*; *R*^2^*_Adj_ = 0.9077*					

**Table 6 polymers-13-03338-t006:** Decomposition temperature of composites at different weight losses from thermogravimetric analysis.

Composite	Decomposition Temperature (°C)				
	2%	5%	10%	20%	30%
OPF-PP	369.8	414.4	434.4	449.7	456.1
OPF-S-PP	283.0	327.3	384.8	449.7	461.4
OPF-S-PP-MAgPP	288.4	331.0	389.3	449.3	460.5

## Data Availability

Not applicable.

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
