# Peer review of "Enhanced Mechanical and Thermal Properties of Modified Oil Palm Fiber-Reinforced Polypropylene Composite via Multi-Objective Optimization of In Situ Silica Sol-Gel Synthesis"

_polymers, 2021, doi:10.3390/polym13193338_

Round 1

Reviewer 1 Report

  1. The manuscript is well organized, and the subject is well presented. The methods used are sound and the presentation and discussion of results is logical.
    The manuscript requires some major revisions to bring it to a level worthy of publication. My recommendations are detailed below:
  2. The current study investigates the characteristics of palm fiber reinforced polypropylene natural composite using silica sol gel. For this, the authors study the mechanical and thermal properties such as tensile properties and thermographic analysis.
  3. The abstract does not read well, first of all the findings given are generic, it is not clear what were the findings in terms of tensile strength and thermal properties. I can see some useful findings are mentioned in the highlights and similar ones should be added in the abstract as well. Therefore, Please consider reviewing the abstract and highlight the novelty, major findings and conclusions.
  4. Before the last paragraph, please consider answering the following question: What is the research gap did you find from the previous researchers in your field? Mention it properly. It will improve the strength of the article.
  5. In the materials and methods sections please add the following: 1) Images of the fabricated samples 2) Images of any equipment, test setup or machines used in the study 3) any standards used for the mechanical or thermal tests (I see you added some already, if any others are missing, please add them too).
  6. For table 3 how many times each run was repeated? If just once, then it is not acceptable, and authors must do at least three runs otherwise the credibility of all the results is weak.
  7. Tables 4 and 5 the authors forgot to add the percentage contribution for each of the input parameters and their interaction.
  8. Again, I am not clear how the authors obtained this ANOVA table without repeating the results, please explain or clarify this issue.
  9. Pure error is so large, this means that the accuracy of the model to predict future trends is low or practically impossible. This is mainly due to low number of runs or having other factors which were not investigated in this work.
  10. Line 265 the authors make serious error here, P-value should be less than 0.05 in order for that factor or the interaction between two or more factors to be significant. However, here the authors appear to be claiming that a higher P-value is a good indication of a factor importance on the measured outputs! Please clarify this issue.
  11. The authors are basing all their critical analysis on the ANOVA/DoE model, which is not always accurate, the authors are supposed to provide a more in depth discussion by analysing the results from the tests.
  12. Again the authors jump to optimization process without conducting proper screening study, the way optimisation studies are done is by first conducting screening studies to eliminate non-significant factors, then the authors should carry a factorial study to determine the linear and non linear interactions in the remaining input factors, then finally the authors can conduct optimisation study using RSM (response surface methodology) or similar optimisation techniques. The authors appears to have skipped the screening, jumped straight into the DoE and optimisation using very small number of test runs, therefore I would say that the results reported here are not credible and the optimisation provide will not be accurate/true for any combination of the input parameters with the range tested.
  13. Line 303 “The presence of silica would slightly” how much is this slightly, please use % or a value to indicate how slightly or how high is something or the change. Using % or numbers is a better way to report something in a scientific manner.
  14. Line 305 “were slightly smaller” same as above.
  15. Conclusion is weak and must be improved, instead of writing one paragraph, the authors are encouraged to add bullet points for each of the sub sections discussed in the results and discussion section.
  16. English is clear and manuscript is organised.

Author Response

Comments and Suggestions for Authors:

(1) The manuscript is well organized, and the subject is well presented. The methods used are sound and the presentation and discussion of results is logical. The manuscript requires some major revisions to bring it to a level worthy of publication. My recommendations are detailed below:

Response: Thanks for review the manuscript and provide valuable comments and suggestions to improve the manuscript. The manuscript is revised following the reviewer's comments and suggestions.

(2) The current study investigates the characteristics of palm fiber reinforced polypropylene natural composite using silica sol gel. For this, the authors study the mechanical and thermal properties such as tensile properties and thermographic analysis.

Response: Thanks for the comment. The main objective this study to determine the influence silica sol-gel modified oil palm fiber as reinforcement to enhance the thermal and mechanical properties of the composites.

(3) The abstract does not read well, first of all the findings given are generic, it is not clear what were the findings in terms of tensile strength and thermal properties. I can see some useful findings are mentioned in the highlights and similar ones should be added in the abstract as well. Therefore, Please consider reviewing the abstract and highlight the novelty, major findings and conclusions.

Response:  Revised the abstract.

(4) Before the last paragraph, please consider answering the following question: What is the research gap did you find from the previous researchers in your field? Mention it properly. It will improve the strength of the article.

Response: Added as suggested.

(5) In the materials and methods sections please add the following: 1) Images of the fabricated samples 2) Images of any equipment, test setup or machines used in the study 3) any standards used for the mechanical or thermal tests (I see you added some already, if any others are missing, please add them too).

Response: Inserted Figure 2 Schematic diagram of the production of OPF and OPF-S conducted through sol-gel process in line 169-170. Standards are also being mentioned for mechanical and thermal tests in line 191 and 202.

(6)  For table 3 how many times each run was repeated? If just once, then it is not acceptable, and authors must do at least three runs otherwise the credibility of all the results is weak.

Response: The experiments were conducted in triplicate.  The statement is added in the section 2.2.

(7) Tables 4 and 5 the authors forgot to add the percentage contribution for each of the input parameters and their interaction.

Response: With due respect to reviewer, data presented in Table 4 and Table 5 are not experimental data. These data are generated by Design expert software for ANOVA analyses, including Sum of squares, degree of freedom, mean squires, F-values and P-values.  No percentage contribution were provided by software.

(8) Again, I am not clear how the authors obtained this ANOVA table without repeating the results, please explain or clarify this issue.

Response:  Thanks for the comment. In Central composite design of experiments, there are 6 centre points. In centre points of experiments of pH values (pH 9), soaking time (22 h) and drying time (75 min), and molar ratio, the data were repeated 6 times. The ANOVA tables are generated from the 6 central points of experiments.

(9) Pure error is so large, this means that the accuracy of the model to predict future trends is low or practically impossible. This is mainly due to low number of runs or having other factors which were not investigated in this work.

Response: It is agreed that pure error for the model is large. However, as clarified in a paragraph before Table 4, the relationships between independent factors and responses are sufficient to be described when the model shows p-value of less than 0.05. In our study, models for the selected responses, which were silica content and silica residue showed p-values of less than 0.05, which indicate that both models are statistically significant. This multi-objective design of experiments was constructed to optimize the responses, with minimum number of experiments, compared to the conventional one-factor-at-one-time experimental works.

 (10) Line 265 the authors make serious error here, P-value should be less than 0.05 in order for that factor or the interaction between two or more factors to be significant. However, here the authors appear to be claiming that a higher P-value is a good indication of a factor importance on the measured outputs! Please clarify this issue.

Response:  Thanks for the correction. We have amended the line 265.

(11) The authors are basing all their critical analysis on the ANOVA/DoE model, which is not always accurate, the authors are supposed to provide a more in depth discussion by analysing the results from the tests.

Response: Revised.

(12)  Again the authors jump to optimization process without conducting proper screening study, the way optimisation studies are done is by first conducting screening studies to eliminate non-significant factors, then the authors should carry a factorial study to determine the linear and non linear interactions in the remaining input factors, then finally the authors can conduct optimisation study using RSM (response surface methodology) or similar optimisation techniques. The authors appears to have skipped the screening, jumped straight into the DoE and optimisation using very small number of test runs, therefore I would say that the results reported here are not credible and the optimisation provide will not be accurate/true for any combination of the input parameters with the range tested.

Response:  Thanks for the suggestions. In the present study, we have utilized the Design expert software (ver.11) to optimize the experimental conditions of pH, soaking and drying times and molar ratio of TEOS to water to maximize the silica content and silica residue. 

(13) Line 303 “The presence of silica would slightly” how much is this slightly, please use % or a value to indicate how slightly or how high is something or the change. Using % or numbers is a better way to report something in a scientific manner.

Response: Corrected as suggested.

(14). Line 305 “were slightly smaller” same as above.

Response: Corrected.

(15) Conclusion is weak and must be improved, instead of writing one paragraph, the authors are encouraged to add bullet points for each of the sub sections discussed in the results and discussion section.

Response: Revised as suggested.

 (16) English is clear and manuscript is organised.

Response: Thanks for the comments. Appreciate the guidance provided by the reviewer.

Reviewer 2 Report

The current manuscript investigates composites made of palm oil fiber, polypropylene and silica nanoparticles aiming at the reuse of available biomass and improvement in the properties of the polymer matrix. The text is short, focusing too much on data presentation with minor discussions. It is not clear the potential use of the composites because of negligible, or little, improvements in the thermal and mechanical properties of the resulting materials. Moreover, the absence of imaging analysis prevents any quantitative conclusion in terms of several important properties including particle size, particle adhesion and dispersion in the polymer matrix, matrix porosity, surface roughness and composite morphology. Based on this, at the current stage, I don't think this work is suitable for publication. The study would be highly benefited by SEM or AFM analysis and extended discussions of the results, making it appropriate for a resubmission in the near future.

Author Response

The current manuscript investigates composites made of palm oil fiber, polypropylene and silica nanoparticles aiming at the reuse of available biomass and improvement in the properties of the polymer matrix. The text is short, focusing too much on data presentation with minor discussions. It is not clear the potential use of the composites because of negligible, or little, improvements in the thermal and mechanical properties of the resulting materials. Moreover, the absence of imaging analysis prevents any quantitative conclusion in terms of several important properties including particle size, particle adhesion and dispersion in the polymer matrix, matrix porosity, surface roughness and composite morphology. Based on this, at the current stage, I don't think this work is suitable for publication. The study would be highly benefited by SEM or AFM analysis and extended discussions of the results, making it appropriate for a resubmission in the near future.

 Response: Thanks for reviewing the manuscript and provide valuable comments and suggestions to improve the manuscript. The authors have been amended the manuscript following the reviewer's comments and suggestions. We amended the discussion of the manuscript. Extensive explanations and results of OPF and OPF-S for SEM imaging and EDX analysis have been added.

Round 2

Reviewer 1 Report

  1. Many figures in the manuscript are not clear and have poor resolution, please fix this issue.
    The authors appear to lack of knowledge about ANOVA, you can calculate the percentage contribution from ANOVA tables, please do some further reading and not just depend on the software, or at least generate a pareto chart to show which of the significant input parameters had the highest contribution. 
    The authors in their answers keep referring to the software, this is not acceptable and some basic knowledge of design of experiments is necessary rather than just mentioning the software and how it works, the authors should conduct more research into how DoE works and what are the different stages in experimental testing using statistical techniques. 

Author Response

  1. Many figures in the manuscript are not clear and have poor resolution, please fix this issue.
    The authors appear to lack of knowledge about ANOVA, you can calculate the percentage contribution from ANOVA tables, please do some further reading and not just depend on the software, or at least generate a pareto chart to show which of the significant input parameters had the highest contribution. 

The authors in their answers keep referring to the software, this is not acceptable and some basic knowledge of design of experiments is necessary rather than just mentioning the software and how it works, the authors should conduct more research into how DoE works and what are the different stages in experimental testing using statistical techniques. 

Response: Thanks for providing valuable comments and suggestions. We have revised the Figures in the manuscript. We have added regression coefficient values in the ANOVA tables. Also, we have revised the regression models and ANOVA tables.   

Reviewer 2 Report

Authors have provided all requested information and made substantial improvemets in the text. The manuscript can be published in the current form.

Author Response

Authors have provided all requested information and made substantial improvemets in the text. The manuscript can be published in the current form.

Response: Thanks for review the manuscript and provided invaluable comments and suggestions to improve the manuscript.
